# The Role of Temperature on the Degree of End-Closing and Filling of Single-Walled Carbon Nanotubes

**DOI:** 10.3390/nano11123365

**Published:** 2021-12-11

**Authors:** Magdalena Kierkowicz, Elzbieta Pach, Julio Fraile, Concepción Domingo, Belén Ballesteros, Gerard Tobias

**Affiliations:** 1Institut de Ciència de Materials de Barcelona (ICMAB-CSIC), Campus UAB, Bellaterra, 08193 Barcelona, Spain; mkierkowicz@gmail.com (M.K.); julio@icmab.es (J.F.); conchi@icmab.es (C.D.); 2Catalan Institute of Nanoscience and Nanotechnology (ICN2), CSIC and the Barcelona Institute of Science and Technology, Campus UAB, Bellaterra, 08193 Barcelona, Spain; epach@icmab.es (E.P.); belen.ballesteros@icn2.cat (B.B.)

**Keywords:** carbon nanocapsules, filled carbon nanotubes, end-closing, encapsulation, sealing

## Abstract

Carbon nanotubes (CNTs), owing to their high surface area-to-volume ratio and hollow core, can be employed as hosts for adsorbed and/or encapsulated molecules. At high temperatures, the ends of CNTs close spontaneously, which is relevant for several applications, including catalysis, gas storage, and biomedical imaging and therapy. This study highlights the influence of the annealing temperature in the range between 400 and 1100 °C on the structure and morphology of single-walled CNTs. The nitrogen adsorption and density functional theory calculations indicate that the fraction of end-closed CNTs increases with temperature. Raman spectroscopy reveals that the thermal treatment does not alter the tubular structure. Insight is also provided into the efficacy of CNTs filling from the molten phase, depending on the annealing temperature. The CNTs are filled with europium (III) chloride and analyzed by using electron microscopy (scanning electron microscopy and high-resolution transmission electron microscopy) and energy-dispersive X-ray spectroscopy, confirming the presence of filling and closed ends. The filling yield increases with temperature, as determined by thermogravimetric analysis. The obtained results show that the apparent surface area of CNTs, fraction of closed ends, and amount of encapsulated payload can be tailored via annealing.

## 1. Introduction

Single-walled carbon nanotubes (SWCNTs) are cylindrical nanostructures composed of individual rolled-up carbon sp^2^ sheets arranged as adjacent hexagons. As-produced SWCNTs are usually closed with hemifullerene caps. The diameter ranges between 0.7 and 3 nm, depending on the method of synthesis. Owing to their high surface area-to-volume ratio, SWCNTs have strong potential for water purification, drug delivery, tissue engineering, catalysis, sensors, and photovoltaics to name some of their applications [1,2,3,4,5]. Different molecules and particles can be attached onto their external surface or inside the tubular channels, thus expanding their functionality.

There is a special interest in using SWCNTs as enclosures for the encapsulation of payloads [6,7,8,9]. The nanotubes offer protection to the hosted material from the external environment, thus allowing, for instance, the encapsulation of air sensitive materials [10], gases [11], and even radionuclides [12,13]. Furthermore, a large variety of unprecedented structures have been observed by confinement of materials at the nanoscale [14]. Filled SWCNTs find applications in diverse areas, including nanoelectronics, magnetic recording, nanobiotechnology, sensors, spintronics, catalysis, energy storage, and thermoelectrics [15]. Bulk filling of SWCNTs in general results in samples containing unwanted material external to the nanotube walls. The presence of external material can dominate the properties of the resulting hybrids and can also induce side effects when employed in the biomedical field. Unless the encapsulated payloads have a strong interaction with the inner nanotube walls, it is necessary to seal/block the ends of the SWCNTs to allow for the selective removal of the non-encapsulated compounds while preserving the inner cargo [16]. SWCNTs can be filled with a chosen payload by using different methods, the most commonly employed being solution phase, molten phase, and sublimation. When the filling process takes place at a high enough temperature, typically a few hundred degrees Celsius, spontaneous closure of the ends of CNTs occurs simultaneously [17]. Fullerenes can also be used to block the ends of both SWCNTs [18,19] and multi-walled CNTs (MWCNTs) [20] in the low-temperature methods. However, this process can be reversible, since the removal of fullerene corks can be triggered by changes in the solvent [19] and pH when the fullerenes are functionalized [21]. Filled and closed-ended CNTs can be referred to as “carbon nanocapsules” [22] because the walls of the nanotubes provide protection to the encapsulated cargo [23].

Closed- and opened-end CNTs have different textural properties that can be monitored by low-temperature N_2_ adsorption [24,25,26,27]. The Brunauer–Emmett–Teller (BET) theory is used here for the calculation of the apparent specific surface area of CNTs [28,29]. CNTs tend to form bundles owing to Van der Waals interactions. Therefore, the potential surfaces available for gas adsorption on SWCNTs include the outer area, inner channels (mainly micropores), interstitial channels, and grooves [30,31], as schematically illustrated in Appendix A (see Supporting Information). Studies comparing the adsorption from a liquid and gaseous phase in CNTs prove that the outer and inner surfaces of the isolated tubes play the most important role in adsorption [32]. Moreover, the presence of impurities, amorphous carbon, and functional groups may reduce the adsorption capacity of CNTs [33]. Therefore, a thermal treatment is sometimes required to restore the storage capacity of CNTs. For instance, annealing is widely used to recover and improve the performance of CNT-based catalysts and sensors [34,35]. However, it should be taken into account that when a sufficiently high temperature is achieved, CNTs end-closure takes place, and their internal cavity is no longer accessible for adsorption. It has been shown that closed-cap nanotubes exhibit high current stability, which is of advantage for their use as electron sources [36]. While the end-closure of SWCNTs by thermal annealing is known to occur [37], the fraction of capped ends at different temperatures, and the consequent implications for filling, remain to be explored.

Herein, we perform a systematic study to understand the role of temperature on the degree of end-closing of SWCNTs and filling yield when carbon nanotubes are filled with EuCl_3_ by molten phase capillary wetting. Europium–CNT hybrids have received attention due to the luminescent properties that europium confers to the system [38,39,40].

## 2. Materials and Methods

Chemical vapor deposition (CVD)-grown SWCNTs (Elicarb^®^, Thomas Swan & Co. Ltd., Consett, UK) contained a mixture of both single-walled (SWCNTs) and double-walled CNTs (DWCNTs). For ease of description, this mixture has been referred to as SWCNTs throughout the manuscript. According to the supplier, the average diameter of the CNTs was 2.1 nm. Iron catalyst particles surrounded by graphitic shells, graphitic particles and amorphous carbon were also present in the sample. Therefore, previous to any processing, CNTs were purified and open-ended with water steam (4 h) and hydrochloric acid (HCl) reflux (6 h), based on a method described in previous studies [41]. This method was employed because no damage to the tubular structure has been reported even after prolonged treatments [42]. Steam reacts through the ends of the carbon nanotubes, preserving the sp^2^ backbone. For each study, the same batch of purified CNTs was used. One batch of purified CNTs (250 mg) was split into five equal fractions. Each sample (50 mg) was sealed in a silica tube under vacuum. One sample was directly opened at room temperature (RT), with no thermal treatment, and referred to as RT-CNTs. The rest of the ampoules, containing the samples under vacuum, were heated at 300 °C min^−1^ in a furnace for 12 h at the selected temperatures of 400, 700, 900, and 1100 °C. After the thermal treatment the ampoules were opened on a laboratory bench (in air).

A second batch of purified CNTs was used for filling with a molten salt. For this, purified and open-ended CNTs were mixed with EuCl_3_ (europium (III) chloride anhydrous, 99.99%, Darmstadt, Alemania) in a weight ratio of 1:10, and then ground using an agate mortar and pestle inside an argon-filled glovebox (Labconco, Kansas City, MO, USA). The sample was split into four equal fractions (125 mg each). Each fraction was introduced into an individual silica tube and sealed under vacuum. The filling experiment was performed using the same program and temperature values as those applied for empty CNTs. After the thermal treatment the ampoules were opened on a laboratory bench (in air). The EuCl_3_-filled CNTs (EuCl_3_@CNTs) contained some unencapsulated material that was removed by extensive washing with hot water.

TGA (Q5000 IR) was performed in air using a heating ramp of 10 °C min^−1^ and up to 900 °C.

Raman spectra were recorded using a LabRam HR8000 (Jobin-Yvon, Palaiseau, France) Raman spectrometer with an excitation wavelength of 532 nm (Ar laser, 10% power). A 100× objective was used in this study. The abscissa was calibrated using a silicon standard. Samples were deposited from an isopropanol dispersion as thin films on glass microscopy slides. The reported I_D_/I_G_ values corresponded to the average of three measurements registered for the same sample at different spots (three accumulations per spot, accumulation time of 5 s). The spectra were normalized to the intensity of the G band.

Nitrogen adsorption isotherms were measured at the temperature of liquid nitrogen (−196 °C) using an ASAP-2020 Micromeritics equipment. Before adsorption, the samples were degassed at 300 °C for 12 h.

High-angle annular dark-field scanning transmission electron microscopy (HAADF-STEM) images were acquired by using scanning electron microscopy (SEM; FEI Magellan XHR) at 20 kV with STEM. High resolution transmission electron microscopy (HRTEM) images were obtained by using FEI Tecnai G2 F20 at 200 kV. Elemental mapping of the sample was performed using an EDAX super ultra-thin window X-ray detector coupled to the Tecnai HRTEM. All samples were drop cast from the dispersion in anhydrous ethanol on lacey carbon Cu grids (Agar Scientific, Stansted, UK).

## 3. Results and Discussion

The as-received, purified (RT) and annealed (at 400, 700, 900, and 1100 °C) empty SWCNTs were studied by conducting low-temperature nitrogen adsorption tests. The apparent specific surface area (S_BET_) was determined by applying the BET method (Figure 1a). Pore size distribution of the different samples was computed using a model based on nonlocal density functional theory (NLDFT). To estimate the cumulative pore volume (V_C_), only the pores in the range of 0.3–1.5 nm were considered. This range was chosen by observing the representation of the NLDFT differential pore volume vs. pore width (Figure 1b). Actually, this range of pore diameters is in agreement with the diameters of the SWCNTs used (containing a fraction of DWCNTs), previously determined by employing Raman spectroscopy and high-resolution transmission electron microscopy (HRTEM) [43]. As expected, the size of the pores is smaller than the actual diameter of the nanotubes. The values were also consistent with the information provided by the manufacturer.

Figure 1 and Appendix A (see Supporting Information) show that the as-received CNTs have a significantly lower S_BET_ and V_C_ than purified SWCNTs (RT). This result shows that the steam treatment efficiently opens the ends of the SWCNTs and removes carbonaceous impurities, thereby significantly increasing the accessible S_BET_ from 791 in as-received to 1246 m^2^ g^−1^ in RT-CNTs and V_C_ from 0.146 to 0.231 cm^3^ g^−1^. The high values of these textural properties were largely retained by annealing up to temperatures of 700 °C. However, annealing of purified CNTs at 900 °C and above resulted in a progressive decrease in S_BET_ and V_C_, although this reduction was less pronounced than the increase observed between the as-received and purified RT-CNTs. This observation indicates that at 900 °C, the ends of the CNTs started to close, and the fraction of the end-closed CNTs increased as a function of temperature.

Raman spectroscopy did not reveal any significant differences between the purified CNTs (RT) and the thermally treated samples. The I_D_/I_G_ values (Figure 2) were similar in all cases and remained between 0.11 and 0.14 (Appendix A; see Supporting Information), indicating that annealing in this range of temperatures did not significantly alter the tubular structure of CNTs. Taking into account the aspect ratio of the employed carbon nanotubes, the I_D_/I_G_ values do not provide information on whether the ends of the nanotubes are opened or closed.

Thermogravimetric analysis (TGA) of the empty and filled SWCNTs was performed under air flow. Figure 3a shows the influence of annealing at selected temperatures on the empty purified CNTs. Compared with the as-received, both the purified (RT) and all the samples of annealed SWCNTs exhibit higher thermal stability against oxidation by air. This is because the steam and HCl treatment removes both carbonaceous impurities and metal catalyst from the sample. The carbonaceous impurities present mainly consist of amorphous carbon and graphitic particles [44]. The presence of such carbonaceous impurities is responsible for the initial weight loss observed at 300–400 °C in the TGA of the as-received SWCNTs, which can be better appreciated in the inset of Figure 3a. The amorphous carbon is much more reactive with oxygen and oxidizes first. The inorganic solid residue collected after the complete combustion of the as-received SWCNTs was 4.6 wt.%. After purification, it decreased to approximately 1.3 wt.%, and remained at this value after the thermal treatment, within experimental error. The as-received sample has a lower combustion temperature than the purified SWCNTs (RT) due to the presence of the carbonaceous impurities in the former. The metal catalyst might also play a role, because samples containing higher concentrations of metal impurities showed a decrease in the temperature required to oxidize the sample [45].

Next, three temperatures (700, 900, and 1100 °C) were employed for the melt filling of purified CNTs with EuCl_3_. The resulting samples were independently washed to remove EuCl_3_ that remained on the exterior of the CNTs [46]. The amount of EuCl_3_ encapsulated after the filling experiment at each temperature was estimated from TGA measurements performed under flowing air (Figure 3b). For EuCl_3_@CNTs samples, carbon oxidizes first, and the remaining residues corresponded to Eu_2_O_3_ formed by oxidation of the metal halide. The formula reported by Ballesteros et al. [47] was used to calculate the amount of encapsulated payload (EuCl_3_) from the TGA residue of each sample. The filling yields turned out to be 3.1, 7.3 and 19.3 wt.% for samples treated at 700, 900 and 1100 °C, respectively. As can be observed, the amount of inorganic residue after the TGA analyses increased with the temperature used in the filling process. Hence, the sample obtained at the highest temperature was characterized by the highest filling yield. Based on these results, it is shown that the filling yield is temperature-dependent, and the amount of encapsulated payload can be controlled by controlling the melt temperature. The higher filling yield obtained at higher temperatures is correlated with the fraction of CNTs with closed ends, i.e., the highest being the number of CNTs having closed ends, the lowest being the probability of washing-out of the encapsulated payload. Nevertheless, it could also be the case that by increasing the temperature used for the filling experiment, EuCl_3_ more easily enters the cavities of CNTs. Temperature plays a key role in filling CNTs; thermal cycles have been previously employed to achieve a high filling yield [48], and it is also known that the temperature of the melt determines the crystal structure of the resulting confined material, for instance, from nanowires to van der Waals heterostructures [49].

After the filling and washing processes, the CNTs were characterized by using electron microscopy. HAADF-STEM was initially employed to assess the overall filling of the sample (Figure 4a). In this imaging modality, the intensity offered by an element is proportional to its atomic number. Therefore, carbon appears as pale gray and the encapsulated material, containing europium, as bright lines following the shape of the carbon nanotube bundles. There are several bright lines corresponding to the filling material. The small bright particles correspond to filling, of either CNTs or graphitic particles, in the form of nanoparticles or to catalytic particles that remain after the purification step. Bundle crossing also results in higher intensity. HRTEM was employed to confirm the absence of non-encapsulated europium chloride, to examine the crystalline structure of the filling, and the condition of the ends after the filling experiment at 1100 °C (Figure 4b and Appendix A; see Supporting Information). As it can be seen in the images, carbon nanotubes are irregularly shaped graphitic particles were found to be filled. The image in Figure 4b shows bundles of CNTs containing both filled and empty nanotubes and some visible closed ends. The presence of empty nanotubes was observed in all the samples. Therefore, it is better to employ TGA to assess the loading efficiency [47]. As a guide to the eye, white arrows point to CNTs filled with nanowires of EuCl_3_, empty white arrows to filling with nanoparticles, and black arrows point to closed ends. Having closed ends prevents leakage of the payload during the washing steps. Notably, during HRTEM analysis a single-layered inorganic nanotube of EuCl_3_ was found inside the cavities of a large diameter DWCNT (see Supporting Information Appendix Ab), thus creating a tubular (1 D) van der Waals heterostructure, as previously observed for other encapsulated 2D materials [50,51,52,53,54]. TEM imaging combined with EDX spectroscopy confirmed the presence of EuCl_3_ in the internal cavity of the CNTs (Figure 4c). Europium and chlorine peaks are clearly visible in the EDX spectrum. The copper peaks arise from the support grid.

It is worth noting that the end-closing of SWCNTs takes place at lower temperatures than MWCNTs [16], and therefore it is a diameter dependent process. It has also been reported that in the case of MWCNTs, the number of walls also plays a role in the degree of end-closing [36]. This is important because whereas the SWNCTs employed in this study have an average diameter of 2.1 nm, SWCNTs from other sources such as arc-discharge, or HiPco typically have smaller diameters. Therefore, lower temperatures than those employed in the present study might be enough to efficiently close the ends of the nanotubes and render a high filling yield.

## 4. Conclusions

The effect of temperature on the degree of end-closing and filling of SWCNTs has been investigated. Low-temperature nitrogen adsorption tests showed that the S_BET_ and Vc of the CNTs increased after purification and decreased by annealing in the temperature range of 900–1100 °C, which is associated with the progressive closure of the ends of the CNTs. Electron microscopy and TGA analysis indicate that annealing did not affect the purity and structure of the CNTs. Using EuCl_3_@CNTs, it was demonstrated that increasing the annealing temperature had a positive effect on the filling yield from the molten phase, which was found to be correlated with the fraction of CNTs with closed ends. In summary, the fraction of end-closed CNTs and the filling yield can be tuned by controlling the annealing temperature.

## Figures and Tables

**Figure 1 nanomaterials-11-03365-f001:**
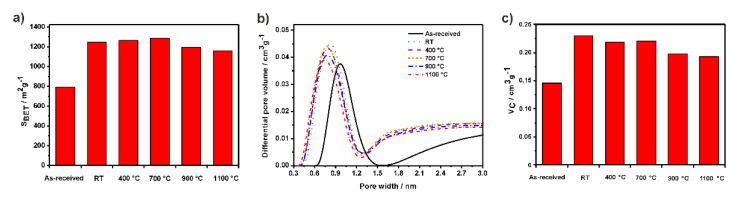
(**a**) Histogram representing the apparent specific surface area (S_BET_), (**b**) nonlocal density functional theory (NLDFT) pore size distribution, and (**c**) histogram of cumulative volume (V_C_) for as−received, purified (RT) and annealed SWCNTs (at 400, 700, 900, and 1100 °C).

**Figure 2 nanomaterials-11-03365-f002:**
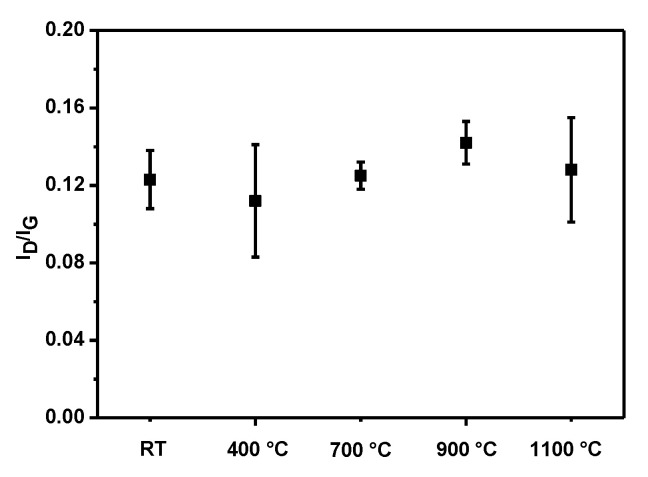
I_D_/I_G_ values extracted from Raman spectra of purified (RT) and annealed SWCNTs (at 400, 700, 900, and 1100 °C). Each value is the average of three measurements. SWCNTs were excited using a laser wavelength of 532 nm.

**Figure 3 nanomaterials-11-03365-f003:**
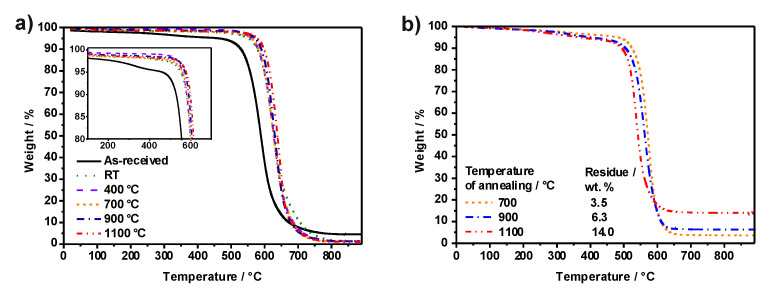
Thermogravimetric analyses of: (**a**) as-received, purified (RT) and annealed SWCNTs (at 400, 700, 900, and 1100 °C); and (**b**) SWCNTs filled with EuCl_3_ at selected temperatures. Measurements were performed under air flow.

**Figure 4 nanomaterials-11-03365-f004:**
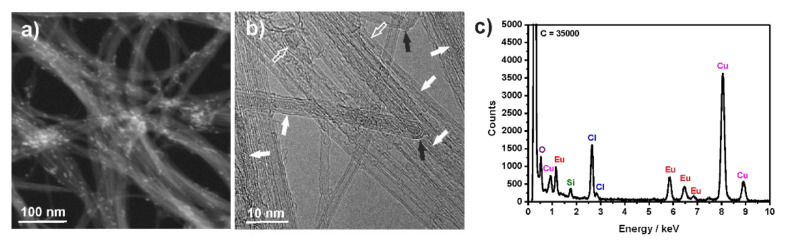
Electron microscopy imaging and energy-dispersive X-ray (EDX) spectroscopy of EuCl_3_@CNTs filled at 1100 °C. (**a**) High-angle annular dark-field scanning transmission electron microscopy (HAADF-STEM) image, (**b**) High resolution transmission electron microscopy (HRTEM) image (as a guide to the eye, white arrows point to CNTs filled with nanowires of EuCl_3_, empty white arrows to filling with nanoparticles, and black arrows point to closed ends) and (**c**) EDX spectrum of the sample, confirming the presence of Eu and Cl (the carbon peak maxima was cut down for better visibility of lower intensity peaks).

## Data Availability

Not applicable.

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
