# Peer review of "The Role of Temperature on the Degree of End-Closing and Filling of Single-Walled Carbon Nanotubes"

_nanomaterials, 2021, doi:10.3390/nano11123365_

Round 1

Reviewer 1 Report

This study highlights the influence of the annealing temperature in the range between 400 and 1100 °C on the structure and morphology of single-walled CNTs. The nitrogen adsorption and density functional theory calculations indicate that the fraction of end-closed CNTs increases with temperature. Raman spectroscopy reveals that the thermal treatment does not alter the tubular structure. Insight is also provided into the efficacy of CNTs filling from the molten phase, depending on the annealing temperature. The CNTs are filled with europium (III) chloride and analyzed by using electron microscopy (scanning electron microscopy and high-resolution transmission electron microscopy) and energy-dispersive X-ray spectroscopy, confirming the presence of filling and closed ends. The filling yield increases with temperature, as determined by thermogravimetric analysis.

  1. It is hard to see the new points from the present paper. What is the possible application of the present study?
  2. Poor citation format in the Introduction section.
  3. Does Figure 1 have the substantial relation with the present study?
  4. Where are Tables S1, S2,…?
  5. Do you finish the experiment in air or vacuum?
  6. Poor quality of Fig. 2. Where are Figs. 2b and 2c?

This paper is unacceptable because of its poor quality.

Author Response

Response to Reviewer 1 Comments

We would like to thank the reviewer for the useful comments provided on our manuscript. Please find below a point-by-point response to all the concerns raised by the reviewer. Changes in the text have been made using track-chages.

Point 1: It is hard to see the new points from the present paper. What is the possible application of the present study?

Response 1: Filled carbon nanotubes find application in a large variety of areas as indicated in the introduction. Thanks to the comment of the referee we have stressed the importance of having closed ends to render a purified sample of filled tubes. The following sentence has been included: “Bulk filling of SWCNTs in general results in samples containing unwanted material external to the nanotube walls. The presence of external material can dominate the properties of the resulting hybrids and can also induce side effects when employed in the biomedical field. Unless the encapsulated payloads have a strong interaction with the inner nanotube walls, it is necessary to seal/block the ends of the SWCNTs to allow the selective removal of the non-encapsulated compounds while preserving the inner cargo.”

Point 2: Poor citation format in the Introduction section

Response 2: We have revised the introduction and added a description of the interest for having closed ends and also highlighting the interest of using EuCl3 as a filler with the corresponding references. The original submission had 45 references and the new version has 54 references.

Point 3: Does Figure 1 have the substantial relation with the present study?

Response 3: The reviewer is correct that Figure 1 does not show scientific results and it has been moved to the Supporting Information. It is now Figure S1.

Point 4: Where are Tables S1, S2,…?

Response 4: These Tables were included as supporting material. To avoid confusion we have added “see Supporting Information” every time that we refer to material in the SI.

Point 5: Do you finish the experiment in air or vacuum?

Response 5: Samples are opened in air after the thermal treatment. This aspect has been clarified in the experimental section.

Point 6: Poor quality of Fig. 2. Where are Figs. 2b and 2c?

Response 6: Thanks for pointing this out. A new Figure 2 has been included (Figure 1 in the present version of the manuscript).

We believe we have properly addressed all the reviewer comments which allow us to enclose an improved version of the manuscript.

Reviewer 2 Report

This paper is very intersting, and investigated the filler loading peroperty of CNT by controlling the annealing temperature. The experiment design seems to be appropriate. Here are the comments from the reviewer:

  1. A major concern is how about the loading efficiency? Figure 5a shows that most of the CNTs are not filled with fillers. The author should provide a few more TEM images.
  2. Authors should make changes for figure 2c, so that all three subfigures are in the same kind of format.

Author Response

Response to Reviewer 2 Comments

We would like to thank the reviewer for the useful comments provided on our manuscript. Please find below a point-by-point response to all the concerns raised by the reviewer. Changes in the text have been made using track-chages.

Point 1: A major concern is how about the loading efficiency? Figure 5a shows that most of the CNTs are not filled with fillers. The author should provide a few more TEM images.

Response 1: HAADF-STEM has been added to provide an overview of the sample and additional TEM images have been included in Figure S2. We have also added arrows in the image pointing to some filled nanotubes and the corresponding discussion in the main text.

Point 2: Authors should make changes for figure 2c, so that all three subfigures are in the same kind of format.

Response 2: Following the suggestion of the reviewer a new Figure 2c has been included (Figure 1c in the present version of the manuscript).

We believe we have properly addressed all the reviewer comments which allow us to enclose an improved version of the manuscript.

Reviewer 3 Report

The manuscript entitled “The role of temperature on the degree of end-closing and filling of single-walled carbon nanotubes” by Kierkowicz et al. reports on influence of annealing temperature on the fractions of the end-closing of carbon nanotubes (CNTs) and the filling of CNTs with EuCl3. The authors prepared as-received CNTs, purified CNTs, purified and annealed CNTs, and purified and EuCl3-filled CNTs. They conducted nitrogen adsorption measurements, Raman scattering spectroscopy, thermogravimetric analysis, transmission electron microscopy, and energy dispersive x-ray spectroscopy. They also performed calculation of pore size distribution based on the nonlocal density functional theory. With these substantial experiments and calculations, the authors concluded that the fraction of CNTs filled with EuCl3 is correlated with the fraction of CNTs with closed ends. The paper provides the details of experiments and results as well as the insight on the end-closing and filling of CNTs. Thus, it has a potential to deserve publication in Nanomaterials. However, before further consideration of publication, I recommend the authors to consider the following concerns.

(1) Figure 3 shows similar ID/IG values for all the samples. The translational symmetry is broken at the end even in the closed-end CNTs. So, I guess the ID/IG estimation cannot suitable to examine whether CNTs have opened or closed ends.

(2) Slightly related to the above concern, could water stream and HCl reflux make entrance holes for molecules on side walls as well as nanotube ends?

(3) Figure 4a shows different temperatures of steep weight losses between the as-received sample and the first batch samples. The as-received sample should involve CNTs, so that steep weight loss for CNTs could be observed at almost the same temperature. Why was the difference observed? Did the present of the metal catalyst make influence on the combustion of CNTs?

- Minor remark

In the experimental section, the description on HAADF-STEM images was present. However, these images were not present in the paper.

Author Response

Response to Reviewer 3 Comments

We would like to thank the reviewer for the useful comments provided on our manuscript. Please find below a point-by-point response to all the concerns raised by the reviewer. Changes in the text have been made using track-chages.

Point 1: Figure 3 shows similar ID/IG values for all the samples. The translational symmetry is broken at the end even in the closed-end CNTs. So, I guess the ID/IG estimation cannot suitable to examine whether CNTs have opened or closed ends.

Response 1: The reviewer is correct in this observation. The aspect ratio of the nanotubes would also not allow to see changes in the ends. We have added a sentence to clarify this aspect.

Point 2: Slightly related to the above concern, could water stream and HCl reflux make entrance holes for molecules on side walls as well as nanotube ends?

Response 2: The steam HCl purification treatment was developed in earlier studies, and no tubular damage was observed, thus making the purified material useful for the permanent containment of materials. A sentence clarifying this aspect has been included in the revised version.

Point 3: Figure 4a shows different temperatures of steep weight losses between the as-received sample and the first batch samples. The as-received sample should involve CNTs, so that steep weight loss for CNTs could be observed at almost the same temperature. Why was the difference observed? Did the present of the metal catalyst make influence on the combustion of CNTs?

Response 3: The difference in combustion temperature is due to the presence of both carbonaceous impurities and metal catalyst in the sample. A sentence clarifying this aspect has been included.

Point 4: Minor remark - In the experimental section, the description on HAADF-STEM images was present. However, these images were not present in the paper.

Response 4: The new version does actually include HAADF-STEM so the description was preserved.

We believe we have properly addressed all the reviewer comments which allow us to enclose an improved version of the manuscript.

Round 2

Reviewer 1 Report

I am happy with the revisions.

Reviewer 2 Report

Publish it as it is.

Reviewer 3 Report

The paper has been improved. It provides the details of experiments and results as well as the insight on the end-closing and filling of CNTs. I think this paper deserves publication in Nanomaterials.